# The Screening of the Protective Antigens of *Aeromonas hydrophila* Using the Reverse Vaccinology Approach: Potential Candidates for Subunit Vaccine Development

**DOI:** 10.3390/vaccines11071266

**Published:** 2023-07-21

**Authors:** Ting Zhang, Minying Zhang, Zehua Xu, Yang He, Xiaoheng Zhao, Hanliang Cheng, Xiangning Chen, Jianhe Xu, Zhujin Ding

**Affiliations:** 1Jiangsu Key Laboratory of Marine Bioresources and Environment, Co-Innovation Center of Jiangsu Marine Bio-Industry Technology, Jiangsu Ocean University, Lianyungang 222005, China; zhangting@jou.edu.cn (T.Z.); zhangminying@jou.edu.cn (M.Z.); xuzehua@jou.edu.cn (Z.X.); zhaoxiaoheng@jou.edu.cn (X.Z.); chenghl@jou.edu.cn (H.C.); xnchen@jou.edu.cn (X.C.); 2005000034@jou.edu.cn (J.X.); 2Jiangsu Key Laboratory of Marine Biotechnology, School of Marine Science and Fisheries, Jiangsu Ocean University, Lianyungang 222005, China; 3Key Laboratory of Sichuan Province for Fishes Conservation and Utilization in the Upper Reaches of the Yangtze River, Neijiang Normal University, Neijiang 641000, China; he_yang_yang@126.com; 4Jiangsu Institute of Marine Resources Development, Lianyungang 222005, China

**Keywords:** *Aeromonas hydrophila*, ERIC-PCR, outer membrane proteins, protective antigens, reverse vaccinology

## Abstract

The threat of bacterial septicemia caused by *Aeromonas hydrophila* infection to aquaculture growth can be prevented through vaccination, but differences among *A. hydrophila* strains may affect the effectiveness of non-conserved subunit vaccines or non-inactivated *A. hydrophila* vaccines, making the identification and development of conserved antigens crucial. In this study, a bioinformatics analysis of 4268 protein sequences encoded by the *A. hydrophila* J-1 strain whole genome was performed based on reverse vaccinology. The specific analysis included signal peptide prediction, transmembrane helical structure prediction, subcellular localization prediction, and antigenicity and adhesion evaluation, as well as interspecific and intraspecific homology comparison, thereby screening the 39 conserved proteins as candidate antigens for *A. hydrophila* vaccine. The 9 isolated *A. hydrophila* strains from diseased fish were categorized into 6 different molecular subtypes via enterobacterial repetitive intergenic consensus (ERIC)-PCR technology, and the coding regions of 39 identified candidate proteins were amplified via PCR and sequenced to verify their conservation in different subtypes of *A. hydrophila* and other *Aeromonas* species. In this way, conserved proteins were screened out according to the comparison results. Briefly, 16 proteins were highly conserved in different *A. hydrophila* subtypes, of which 2 proteins were highly conserved in *Aeromonas* species, which could be selected as candidate antigens for vaccines development, including type IV pilus secretin PilQ (AJE35401.1) and TolC family outer membrane protein (AJE35877.1). The present study screened the conserved antigens of *A. hydrophila* by using reverse vaccinology, which provided basic foundations for developing broad-spectrum protective vaccines of *A. hydrophila*.

## 1. Introduction

*Aeromonas hydrophila*, a rod-shaped Gram-negative bacterium, is an important member of the genus *Aeromonas* of *Vibrionaceae*. It is conditionally pathogenic and commonly found in nature. The pathogenic *A. hydrophila* can infect various species of fish and cause explosive necrotizing fasciitis, resulting in enormous economic losses to the aquaculture industry [1,2,3]. The typing of potentially pathogenic *A. hydrophila* is an important means of clinical epidemiological investigation, which can also provide a scientific basis for cutting off the transmission route and controlling the outbreak of related diseases, and also has certain guiding significance for the preparation and application of more targeted vaccines. It has been found that *A. hydrophila* shows high intraspecific genetic diversity via the analysis using random amplified polymorphic DNA (RAPD) and enterobacterial repetitive intergenic consensus (ERIC)-PCR markers [4]. The distribution of virulence-related genes also indicates that *A. hydrophila* is a genetically heterogeneous species with different pathogenicity potential for humans and other animals [5]. The main methods to control and treat diseases caused by *A. hydrophila* infection are antibiotic treatment or vaccine immunization. However, the problem of strain resistance affects the effectiveness of antibiotic usage, and the existence of various serotypes also limits the protective effects of vaccines [6].

The outer membrane proteins (OMPs) of Gram-negative bacteria are important virulence factors, which are related to the pathogenicity of bacteria [7]. Because of their high conservation and antigenicity, they have been widely studied as candidate antigens for bacterial vaccines [8]. With the development of bioinformatics, vaccinology, and immunology, novel *A. hydrophila* vaccines with OMPs as candidate antigens became a research hotspot and have been immunologically evaluated on various fish species [9,10]. Wang et al. found that outer membrane protein 38 can effectively stimulate specific and non-specific immune responses after immunizing Chinese breams, and can be used as a potential vaccine antigen for *A. hydrophila* [11]. Abdelhamed et al. used genomic subtraction to identify three proteins found only in higher virulence strains: the major outer membrane protein A1 (OmpA 1), TonB-dependent receptor (Tdr), and transferrin-binding protein A (TbpA), in which OmpA 1 and Tdr proteins provide strong protection against catfish [12]. The outer membrane protein W (OmpW) of *A. hydrophila* was found to be able to prevent *A. hydrophila* infection in *Labeo rohita* via oral immunization [13]. These results indicate that conserved OMPs are often used as candidate antigens for vaccine development in *A. hydrophila* [14]. 

The screening of antigenic proteins is the key approach to preparing a broad-spectrum protective vaccine against *A. hydrophila*. Traditional methods rely on empirical the screening of a few candidate vaccines according to the characteristics of known pathogens. Reverse vaccinology uses high-throughput screening of the whole genome of pathogens to identify genes encoding proteins that exhibit excellent immunogenicity, and then the recombinant proteins can be detected in vitro or in vivo. Reverse vaccinology has been applied to a variety of pathogens, including *Neisseria meningitidis*, *Streptococcus agalactis*, *Streptococcus pyogenes*, *Streptococcus pneumoniae,* and *Escherichia coli* [15]. Previously, a total of 14 proteins have been identified as potential candidate vaccine antigens from the whole cell biofilm of *A. hydrophila* by using the reverse vaccinology method, which can protect the experimental fish from the infection of different geographical isolates of *A. hydrophila* [16]. 

In this study, candidate antigens for the development of *A. hydrophila* vaccines were screened by performing signal peptide prediction, transmembrane helical structure prediction, subcellular localization, antigenicity, and adhesion evaluation, as well as interspecific and intraspecific homology comparison of 4268 encoded protein sequences in *A. hydrophila* J-1 through the use of reverse vaccinology. The categorization of typically identified *A. hydrophila* strains from diseased fish was accomplished with ERIC-PCR technology, and the conservation of the selected candidate proteins in different subtypes of *A. hydrophila* was verified by amplifying them through PCR and sequencing. Furthermore, the candidate protein sequences were compared with those of *Aeromonas* species, including *Aeromonas caviae*, *Aeromonas veronii*, *Aeromonas salmonicida*, and *A. hydrophila*, and conserved proteins were screened out based on the results of the comparison. This study represents a preliminary step towards screening candidate antigens for the development of a broad-spectrum vaccine against *A. hydrophila* and will provide the foundation for the therapy and control of diseases resulting from *A. hydrophila* infection.

## 2. Materials and Methods

### 2.1. Ethics Statement

This study was approved by the Animal Care and Use Committee of Jiangsu Ocean University (protocol no. 2020-37, approval date: 1 September 2019). All animal procedures were performed in accordance with the Guidelines for the Care and Use of Laboratory Animals in China.

### 2.2. Bioinformatic Analysis of the A. hydrophila Genome

In the present study, reverse vaccinology was applied to provide the basis for the preparation of subunit vaccines by predicting the signal peptides, transmembrane helices, and subcellular localization of candidate proteins (Figure 1).

#### 2.2.1. Data Collection

The whole genome nucleotide acid coding sequences of the *A. hydrophila* J-1 strain were downloaded in the FASTA file format from the NCBI database for a total of 4268 sequences (accession number: CP006883.1).

#### 2.2.2. Signal Peptide Prediction

Using SignalP 5.0 (www.cbs.dtu.dk/services/SignalP accessed on 14 December 2021) online software, the 4268 protein sequences of the *A. hydrophila* J-1 strain were analyzed to predict the presence of signal peptides and the location of hydrolysis sites with Gram-negative prokaryotes as the default setting. 

#### 2.2.3. Transmembrane Helical Structure Prediction 

TMHMM 2.0 (http://www.cbs.dtu.dk/services/TMHMM/ accessed on 16 December 2021) and HMMTOP (http://www.enzim.hu/hmmtop/ accessed on 16 December 2021) online software were used to predict the transmembrane helical structures of *A. hydrophila* J-1 strain-encoding proteins. The lower the number of transmembrane helices in the protein, the less difficult it was for the recombinant protein to be expressed and prepared. 

#### 2.2.4. Prediction of the Subcellular Localization 

To improve the accuracy of the subcellular localization of candidate proteins, the subcellular location of proteins with less than two TM (transmembrane) helices were predicted on the predictive websites, including PSORTb, CELLO, and Gneg-mPLoc, respectively. PSORTb (https://www.psort.org/ accessed on 16 December 2021) is a web-based tool that provides links to the PSORT series of programs to predict subcellular localization. Based on comprehensive analysis, CELLO (http://cello.life.nctu.edu.tw/ accessed on 16 December 2021) can detect specific amino acid composition and predict the subcellular location of the target proteins. Gneg-mPLoc (http://www.csbio.sjtu.edu.cn/bioinf/Gneg-multi/ accessed on 16 December 2021) can analyze the subcellular localization on Gram-negative bacterial coding proteins according to the information of gene ontology, functional domains, and sequence evolution. 

#### 2.2.5. Antigenicity and Adhesion Index Analysis of the TARGET Proteins

Antigenicity and adhesion are important factors to consider when evaluating potential antigens. When bacteria invade, their adhesion structures encounter the immune system of the host. Therefore, immunity against the adhesion structure can help prevent further colonization and infection. The potential antigenicity of candidate proteins was assessed with a cutoff value set to 0.4 with VaxiJen (http://www.jenner.ac.uk/VaxiJen accessed on 17 December 2021). Candidate vaccine antigens were selected from the adhesion proteins with an adhesion index greater than 0.5, which were assessed using Vaxign (http://www.violinet.org/vaxign/ accessed on 17 December 2021). Vaxign is a web-based system for investigating vaccines. Using Vaxitope, the system accurately predicts MHC class I and II binding epitopes, identifies both known targets, and suggests potential candidates. Proteins with an antigen score of 0.5 or higher are more likely to be recognized by the host immune system, making them strong vaccine candidates for further analysis.

#### 2.2.6. Homology Analysis of the Protein Sequences 

To assess the homology of candidate proteins in the genus *Aeromonas*, sequence alignment of the candidate proteins from the *A. hydrophila* J-1 strain with that of 37 *Aeromonas* species (from NCBI database) was performed by using BLAST, including *A. allosaccharophila*, *A. bestiarum*, *A. caviae*, *A. dhakensis*, *A. encheleia*, *A. eucrenophila*, *A. fluvialis*, *A. jandaei*, *A. lusitana*, *media, A. piscicola*, *A. popoffii*, *A. rivipollensis*, *A. salmonicida*, *A. sanarellii*, *A. sobria*, *A. taiwanensis*, *A. tecta*, *A. veronii*, *A. aquatica*, *A. aquatilis*, *A. australiensis*, *A. bivalvium*, *A. cavernicola*, *A. crassostreae*, *A. diversa*, *A. enterica*, *A. enteropelogenes*, *A. finlandensis*, *A. lacus*, *A. guangheii*, *A. hydrophila*, *A. intestinalis*, *A. molluscorum*, *A. rivuli*, *A. schubertii*, and *A. simiae*. 

### 2.3. Conservative Analysis of Candidate Proteins in Different Subtypes of A. hydrophila and Aeromonas Species

#### 2.3.1. Sample Collection

The extracted genomic DNA from 10 strains was used as template DNA for the molecular characterization and ERIC-PCR molecular typing of *A. hydrophila* strains. Detailed information about all used *A. hydrophila* strains in this study is listed in Table 1. A total of 10 suspected *A. hydrophila* strains isolated from five fish, including *Parabramis pekinensis*, *Megalobrama amblycephala*, *Carassius auratus gibelio*, *Ictalurus punctatus*, and *Carassius auratus* were used during this study. Six strains (Ahp001, Ahp002, Ahp004, Ahp003, Ahm001, and Avc001) were isolated and stored by our laboratory, and three strains (Ahx001, Ahx002, and Ahi001) were isolated and stored by Neijiang Normal University (Sichuan Provincial Key Laboratory for the Conservation and Utilization of Fish Resources in the Upper Reaches of the Yangtze River, Neijiang, China), and the Ahc001 strain was purchased from the national strain resource center of Wuhan University. All strains were stored in 50% glycerol at −80 °C until use. All strains were activated in LB medium, and cultivated for 8 h at 37 °C, and the bacterial DNA was extracted using a TIANamp Bacteria DNA Kit (TIANGEN, Beijing, China) according to the manufacturer’s instructions. The extracted DNA was kept at −20 °C until use.

#### 2.3.2. Molecular Characterization of the *A. hydrophila* Strains 

To characterize and verify the collected *A. hydrophila* strains, a pair of primers (F: 5′-AGAGTTTGATCATGGCTCAG-3′, R: 5′GGTTACCTTGTTACGACTT-3′) were used to amplify the conserved region of the *A. hydrophila 16S rDNA* gene [17]. All the reagents added to the PCR premixed solution were obtained from the PCR kits (Takara, Shiga, Japan). The volume of the PCR reaction mixture was 20 μL, including: 2 μL of 10× buffer, 0.5 μL of the dNTP mix, 0.3 μL of each primer, 0.3 μL of Taq DNA polymerase, 1 μL of template DNA, and then the volume was adjusted to 20 μL with ddH_2_O. The PCR reaction procedure was as follows: initial denaturation at 94 °C for 5 min, followed by 35 cycles of denaturation at 94 °C for 30 s, annealing at 52 °C for 30 s, extension at 72 °C for 1 min, and final extension at 72 °C for 10 min. All PCR products were detected with 1% agarose gel and then sequenced (Tsingke, Nanjing, China). Finally, the full-length sequences were assembled using DNAStar software (http://www.dnastar.com accessed on 8 April 2022) and sequence alignment was performed by BLAST. 

#### 2.3.3. ERIC-PCR Analysis of *A. hydrophila* Strains

A pair of universal primers (F: 5′-ATGTAAGCTCCTGGGGGATTCAC-3′, R: 5′-AAGTAAGTACTGGGGGGTGAGCG-3′) were used in the ERIC-PCR assay, which was performed in triplicate to analyze the fingerprinting patterns of all *A. hydrophila* strains [4]. The final volume of the ERIC-PCR mixture was 20 μL, including: 2 μL of 10× buffer, 0.5 μL of dNTP mix, 1 μL of each primer, 0.3 μL of Taq DNA polymerase, 2 μL of template DNA, and the final volume was adjusted to 20 μL with ddH_2_O. The ERIC-PCR reaction procedure was set up as follows: the temperature of pre-denaturation was 94 °C for 5 min, then denaturation at 94 °C for 1 min, annealing at 52 °C for 1 min, and extension at 68 °C for 8 min by 30 cycles. The last extension was set to 65 °C for 16 min. The amplified ERIC-PCR products were detected using electrophoresis on 1.5% agarose gel.

#### 2.3.4. Cluster Analysis of *A. hydrophila* Strains

Quantity One software (Bio-Rad Laboratories, Inc., Hercules, CA, USA) was used to analyze the ERIC-PCR fingerprints of *A. hydrophila* strains, specifically including generation of the electrophoresis diagram, calculating the molecular weight of each band according to the DNA markers, and recording the amplified bands of each sample with ‘1′ when the band existed or ‘0’ when it did not exist. NTSYS-pc software (version 2.1 K, Applied Biostatistics, Inc., New York, NY, USA) was used to analyze the genetic similarity matrix, pedigree, and distance matrix, and the unweighted paired bucket average (UPGMA) method was used to cluster the fingerprints. Those with similarity ≥0.8 were assumed to be the same subtype, while those with a similarity <0.8 were different genotypes. These software have been widely used in the analysis of gel bands and the molecular typing of species [18,19]. 

#### 2.3.5. PCR Amplification of the CDS Regions of Candidate Proteins

The conservation of candidate proteins in different *A. hydrophila* subtypes was detected via PCR amplification of protein-coding sequence (CDS) regions. Starting from the CDS region of the protein, 210 bases were counted both forward and backward. Primers for 39 proteins were designed using Primer Premier 5.0 to ensure amplification of the intact CDS region (Appendix A). The final volume of the PCR mixture was 25 μL, including: 2 μL of 10× buffer, 0.5 μL of dNTP mix, 0.5 μL of each primer, 0.3 μL of ExTaq DNA polymerase, 1 μL of template DNA, and the final volume was adjusted to 25 μL with ddH_2_O. The PCR reaction procedure is set up as follows: The first step involved predenaturation with a temperature set to 94 °C for 5 min. The second step was to perform 30 cycles of denaturation, annealing, and extension, namely denaturation at 94 °C for 1 min, annealing at 56 °C for 1 min, and extending for 2 min at 72 °C; the last extension was set to 72 °C for 10 min. The PCR products were tested for the presence or absence of specific bands with 1% agarose gel electrophoresis and then sequenced (Tsingke). Finally, the full-length sequences obtained from sequencing were analyzed and aligned with those of the *Aeromonas* species in the NCBI database using BLAST.

#### 2.3.6. Sequence Alignment Using BLAST

Twenty-four strains of *Aeromonas* species were selected from the NR database of NCBI (Appendix A). DNAStar software was used to analyze the full-length sequence of sequenced strains and to find open reading frames (ORFs). The ORFs of the sequenced strains was aligned using BLAST against the 39 protein sequences of *A. hydrophila* J-1. The candidate protein sequence of *A. hydrophila* J-1 was compared with the *Aeromonas* species in the NCBI-nr databas, such as *A. caviae*, *A. salmonicida*, *A. veronii*, and other *A. hydrophila* using BLAST. Then, only the protein sequence consistency data with a coverage ratio greater than 80% were retained.

### 2.4. Statistical Analysis

All data are presented as mean ± SEM and statistical significance was assessed with one-way analysis of variance (ANOVA) using SPSS (version 17.0, Chicago, IL, USA), with *p* < 0.05 considered as a statistically significant difference.

## 3. Results

### 3.1. Bioinformatic Analysis of the A. hydrophila Genome

#### 3.1.1. Signal Peptide Prediction

In the present study, reverse vaccinology was applied to provide the basis for the preparation of subunit vaccines by predicting the signal peptides, transmembrane helices, and subcellular localization of candidate proteins (Figure 1). The *A. hydrophila* J-1 strain genome encoded 4268 proteins in total, and 678 proteins with signal peptides in the N-terminal regions. Among them, the number of proteins that possessed common secretory pathway (Sec) signaling peptides, twin-arginine translocation (TAT) signaling peptides, and lipoprotein signaling peptides were 479, 26, and 173, respectively.

#### 3.1.2. Transmembrane Helix Prediction

TMHMM2.0 and HMMTOP analysis showed that 466 proteins had no transmembrane protein α-helix and 158 proteins had one α-helix, which indicated that the structures were mainly β-barrel. β-barrel proteins are mainly porins, which usually exist in the outer membranes of prokaryotes and the organelle membranes of eukaryotes, thereby providing permeability for the outer membrane, participating in the transport of metabolites and proteins, and maintaining the stability of the outer membrane structures. In the prokaryotic expression system, multiple transmembrane helix structures will affect the recombinant expression of proteins; thus, 54 proteins with multiple transmembrane helix structures were excluded, and the remaining 624 proteins could be further analyzed (Appendix A).

#### 3.1.3. Prediction of the Subcellular Localization

The pie chart indicates the prediction results of the three software, namely the specific location of the 624 candidate proteins in the cell. The PSORTb software predicts that about 10.6% of the candidates are outer membrane proteins (Figure 2A). The CELLO software predicts that about 19.9% of the candidate proteins are outer membrane proteins (Figure 2B). The Gneg mPLoc software predicts that about 13.3% of the candidates are outer membrane proteins (Figure 2C). To balance the predicted coverage and confidence, the final screened outer membrane proteins need to meet the conditions of at least two prediction software. Therefore, 66 outer membrane proteins were finally selected (Figure 2D).

#### 3.1.4. Antigenicity and Adhesion Index Prediction of the Candidate Proteins

The antigenicity of a candidate antigen to evaluate the ability of the antigen to be recognized by the host immune system was predicted. The potential antigenicity of the candidate proteins was assessed using VaxiJen with the cutoff value set to 0.4, which showed that 64 of 66 proteins scored higher than the critical value (>0.4), indicating that most of the identified outer membrane proteins were potential antigenic proteins. In addition, the adhesion of outer membrane proteins is related to bacterial pathogenicity, determining whether bacteria can adhere and colonize specific sites, which is also a vital indicator in antigen screening. Vaxign was used to predict candidate vaccine antigens based on the criteria of an adhesion index above 0.5, which showed that 39 of 66 outer membrane proteins met the standard and could be selected as candidate antigens for the development of subunit vaccines.

#### 3.1.5. Homology Analysis of the Candidate Proteins

In order to analyze the homology of candidate proteins in *Aeromonas* species, the candidate protein sequences were compared with 39 other *Aeromonas* genera via BLAST alignment. In total, 15 of 66 candidate proteins showed high conservation in *Aeromonas*, with a percent identity greater than 85% and query coverage greater than 40% (Table 2), including three proteins associated with the β-barrel assembly machinery (BAM) complex: BamA, BamE, and BamC (AJE37434.1, AJE37120.1, and AJE37742.1), outer membrane protein OmpK (AJE37487.1), porin OmpA (AJE37342.1), TolC family outer membrane protein (AJE35877.1), maltoporin LamB (AJE37449.1), TonB-dependent siderophore receptor (AJE38562.1, AJE36112.1, and AJE37629.1), immune inhibitor A (AJE35148.1), glycine zipper 2TM domain-containing protein (AJE38071.1), carbohydrate porin (AJE35595.1), type IV pilus secretin PilQ (AJE35401.1), and porin (AJE38163.1). These proteins play an important role in protein secretion, transport, and signal transduction, and participate in the absorption of iron via bacteria, thus meeting the survival and reproductive requirements of the *Aeromonas* genus.

### 3.2. Molecular Identification of Strains

The size of the 16S rDNA genes sequence was about 1500 bp. These sequences were reversed, aligned, and compared with similar database sequences. BLAST analysis showed that nine strains had high sequence similarity with *A. hydrophila* (99%), and one strain had high sequence similarity with *A. veronii* (99%). The result was consistent with Table 1. 

### 3.3. ERIC-PCR

#### 3.3.1. ERIC-PCR Typing of *A. hydrophila*

The gel electrophoresis map showed that the 10 strains could obtain 3–5 bands with molecular weight between 300 and 5000 bp through ERIC-PCR amplification, indicating gene polymorphism. Most strains had obvious main bands at 1000–1500 bp and 2000–3000 bp (Figure 3A). The result of fingerprint cluster analysis shows that 10 strains of bacteria are divided into 7 genotypes (Figure 3B). Among them, the number of strains of genotype III is relatively large, including three strains.

#### 3.3.2. Cluster Analysis of ERIC-PCR Fingerprints

The fingerprint cluster analysis showed that 10 strains were divided into seven genotypes (Figure 3B). Nine of the *A. hydrophila* strains were classified into six genotypes. Type I strains had the largest number with three strains, two of which were isolated from different ecological niches (kidney and liver) of the same geographical location (Nanjing) and the same fish source (*P. pekinensis*). There are two genotypes isolated from different geographical locations (Huai’an and Wuhan) and different fish sources (*P. pekinensis* and *M. amblycephala*). Each genotype (Ⅲ-Ⅶ) has one strain; the strains were isolated from different geographical locations, fish sources, and ecological niches.

### 3.4. Conservative Analysis of Candidate Proteins in Aeromonas and Different Subtypes of A. hydrophila 

NCBI BLAST was used to compare the sequence of 39 candidate proteins of *A. hydrophila* J-1. It was found that 39 proteins were widely distributed in *A. hydrophila*, with high sequence consistency and good conserved properties. The 24 strains in the comparison results were selected to record the sequence coverage and consistency, and the consistency data with sequence coverage greater than 80% were retained (Figure 4). The results showed that the positive presence rate of 19, 26, 7, and 29 proteins in six strains of *A. caviae*, six strains of *A. salmonicida,* and six strains of *A. veronii* were 100% and the sequence consistency was >80% (Table 3). In *A. veronii*, 32 proteins showed positive presence, but the protein sequence consistency was low; only seven proteins had sequence consistency >80%. These results indicated that 39 proteins had higher conserved values in *A. caviae*, *A. salmonicida,* and *A. hydrophila*, but lower conserved values in *A. veronii*. Among the six strains sequenced, only 16 proteins had a positive presence rate of 100% and sequence consistency >80%, which was significantly different from the six strains of *A. hydrophila* in the NCBI database. The reason for this result may be that some technical limitations failed to obtain the sequence of some proteins or that the strains used for sequencing themselves belonged to different subtypes and were isolated from different fish sources or geographical locations with high heterogeneity. In summary, 16 proteins are highly conserved in *A. hydrophila*, they are outer membrane beta-barrel protein (AJE34708.1), type IV pilus secretin PilQ (AJE35401.1), TolC family outer membrane protein (AJE35877.1), outer membrane protein transport protein (AJE36247.1), OmpA family protein (AJE37343.1), porin OmpA (AJE37342.1), outer membrane protein OmpK (AJE37487.1), TonB-dependent hemoglobin/transferrin/lactoferrin family receptor (AJE37629.1), TonB-dependent siderophore receptor (AJE38562.1), LPS assembly protein LptD (AJE37705.1), porin (AJE37778.1), peptidoglycan DD-metalloendopeptidase family protein (AJE37804.1), TIGR04219 family outer membrane beta-barrel protein (AJE34989.1), immune inhibitor A (AJE35148.1), and DUF2860 family protein (AJE37735.1). Among them, the sequence consistency of two crucial proteins, type IV pilus secretin PilQ (AJE35401.1) and TolC family outer membrane protein (AJE35877.1), was found to be more than 80% across 30 *Aeromonas* strains. This indicates that PilQ and TolC are highly conserved in *Aeromonas*, and suggests that they may play important roles in the biology and pathogenesis of this bacterial genus.

## 4. Discussion

The disease caused by *A. hydrophila* affects many kinds of fish, and the explosive bacterial septicemia caused by its infection results in serious economic losses to aquaculture. The classification system of *A. hydrophila* has been very confusing, which has brought great inconvenience to disease control and scientific research exchange. The molecular typing of 120 *Aeromonas* strains isolated from fecal samples and environmental samples from gastroenteritis patients using RAPD and ERIC-PCR revealed high genetic heterogeneity between these *Aeromonas* species. ERIC-PCR was shown to be useful for epidemiological investigations and population genetic analysis of *Aeromonas* and has the same discriminatory power as the RAPD method [20].

In this study, nine strains of *A. hydrophila* were analyzed using ERIC-PCR. The sample size was small and the prevalence of the strains was complex. Results showed that *A. hydrophila* strains from the same fish source tended to cluster into one genotype. For example, three *A. hydrophila* strains isolated from *P. pekinensis* were classified as genotype Ⅰ. However, strains collected from different ecological niches within the same region did not always cluster into one group, such as strains of genotype II and III that were separated from the gill and liver of *P. pekinensis*. Additionally, strains from different regions and ecological niches could cluster into the same genotype, as observed with strains collected from *P. pekinensis* in Huai’an and *M. amblycephala* in Wuhan classified as the second genotype. This suggests that the genotype of *A. hydrophilic* may be related to fish source, geographical location, and ecological niche, although it is not absolute. This is consistent with the conclusion reached by Aguilera-Arreola et al. that *A. hydrophila* is a genetically heterogeneous species [5]. Cross-regional transmission may be a primary contributor to the observed genetic diversity within strains due to factors like natural water flow and the spread of reproductive products.

The reverse vaccinology approach was first used to identify candidates for vaccines against the serogroup *B N. meningitidis* [21]. Subsequently, reverse vaccinology approaches were widely used. Leow used reverse vaccinology to identify five OMPs against *Shigella flexneri* as potential vaccine candidates [22]. Using signal peptides, transmembrane helices, and subcellular localization bioinformatics prediction, Wang systematically screened 831 OMPs from 1014 non-redundant proteins of *Vibrio parahemolyticus*, and conducted protein homology analysis in 32 species of *Vibrio* spp. According to the positive presence rate of at least 15 species of *Vibrio* spp, the proteins with high homology were selected as potential multivalent *Vibrio* vaccine candidates [23]. In this study, bioinformatics analysis of 4268 protein sequences encoded by *A. hydrophila* J-1 strain whole genome was performed based on reverse vaccinology. The specific analysis including signal peptide prediction, transmembrane helical structure prediction, subcellular localization prediction, and antigenicity and adhesion evaluation, as well as interspecific and intraspecific homology comparison, thereby screened the 39 conserved proteins as candidate antigens for the *A. hydrophila* vaccine. The conserved properties of 39 proteins were evaluated with the PCR amplification of genes from *A. hydrophila* isolates, and the positive presence rate of 16 proteins was 100%, indicating a high coverage rate in sequencing strains. However, the low prevalence of vaccine candidates in circulating strains may result in low coverage.

Many studies have shown that OMPs, as a potential candidate vaccine antigen of bacteria, has the advantages of strong immunogenicity and high safety. Novel OMPs as immunogens were prepared against different subtypes of *A. hydrophila*, which are of great significance for the healthy green cultivation of freshwater fish. In order to identify candidate proteins for vaccine development, bioinformatics analysis was performed, starting with the prediction of the signal peptide. Most secreted proteins of bacteria carry a short peptide at their N-terminal, called the signal peptide [24]. Predicted signal peptide helps to determine the subcellular localization of this protein and to understand the secretion mechanism of the extracellular proteins of this bacterium [25,26]. Our results show that a total of 678 proteins in 4268 proteins of *A. hydrophila* J-1 have signal peptides in the N-terminal region, identified as three signal peptides, including the secretory pathways (Sec) signaling peptides, twin-arginine translocation (TAT) signaling peptides, and lipoprotein signal peptides. Prokaryotes export unfolded proteins from the cytoplasm through the general secretory Sec system [27]. TAT signaling peptides and lipoprotein signal peptides play an important role in the maturation of bacteria [28,29]. To ensure greater accuracy, two software were used to predict protein transmembrane helices, with TMHMM showing a 97% accuracy rate [30]. To cross-check the subcellular location of candidate proteins, three parallel prediction tools were used, including PSORTb and Gneg-mPLoc. The use of PSORTb makes the results more accurate because it is the most accurate prediction tool for bacterial localization [31], and Gneg-mPLoc improves the quality of the subcellular localization [32].

The homology analysis of the *A. hydrophila* OMPs candidates using BLAST revealed a relatively conserved OMP with the other *Aeromonas* species, which provides insight into cross-protection with these OMPs. Sixteen broadly conserved OMPs were identified in the *A. hydrophila* genus based on the overall homology analysis. Many of the conserved proteins are involved in crucial functions. For example, fadl-mediated LCFA and its derivatives play important roles in phospholipid biosynthesis, membrane permeability, and bacterial pathogenicity [33]. TolC family proteins are the channel proteins of Gram-negative bacteria, which play a common role in the expulsion of diverse molecules from the cell [34]. The TonB-dependent receptors are mainly responsible for the recognition and transportation of macromolecular nutrients, such as iron uptake, and play an important role in the virulence of pathogenic bacteria [35,36]. Outer membrane protein A (OmpA) is a highly conserved transmembrane structural protein, which exists in large numbers in the outer membrane of Gram-negative bacteria. It has become a vaccine candidate protein due to its involvement in bacterial biofilm formation, antibiotic resistance, and immune regulation [37,38]. The BAM complex is composed of one transmembrane protein BamA and four lipoproteins BamB, BamC, BamD, and BamE. It is related to the folding and insertion of outer membrane proteins and is the final and key participant in the assembly of OMP [39]. Our results show that the three protein sequences described as BamA, BamC, and BamE were widely found in six genotypes of *A. hydrophila*, and the homology of the protein sequences was more than 80%. BamA is the important center component of the complex, and is a member of the Omp85 superfamily [40]. It contains the central membrane domain that is important for the insertion enzyme function of BAM complex, and is conservative in mitochondria and chloroplasts [41]. BamC is not an important protein in the BAM complex and the gene sequence is less conserved [42]. BamE has high conservatism, and it is speculated that this protein may play a role in mediating BAM complexes or binding to BAM complex substrates through comparison with proteins with similar structures [43]. There are few studies on the t2ss component of *A. hydrophila* Gspd family. In *Klebsiella pneumoniae*, the t2ss component Gspd-k is a relatively stable protein containing signal peptide sequences. It is a secretory protein and exists on the extracellular membrane with a conservative sequence [44]. Type IV pili are associated with bacterial virulence and infection rates, and multifunctional fibers formed on the surface by many Gram-negative bacteria promote bacterial adhesion to the host cells [45].

In past studies, Basmeet identified 14 proteins including TolC, BamA, LamB, and AH4AK4_2542 as whole-cell *A. hydrophila* biofilm vaccine candidates using reverse vaccinology [16]. Among them, these proteins like TolC, BamA, and Lamb were also predicted to be OMPs in this study, and importantly, we confirmed the identity of these OMPs in *Aeromonas* at the sequence level. Additionally, it was found in this study that six OMPs (TolC, BamA, BamC, BamE, OmpK, and OmpA), which are highly conserved across *Aeromonas* spp., could become effective vaccines against *A. hydrophila* infection due to their strong immunogenicity and cross-protective potential against different fish pathogens. TolC is evolutionarily conservative among *Aeromonas*, has strong immunogenicity and antigenicity, and may have cross-immune protection against different fish pathogens [46]. *A. hydrophila* has three TonB energy systems (TBDTs) and they are all involved in the toxic effects of bacteria in zebrafish. With its protective immunogenicity and widespread distribution, tbdt is an excellent candidate for vaccine development [23]. BamA (NGO1801) and LptD (NGO1715) are highly conserved in *Neisseria gonorrhoeae* strains and can serve as potential *N. gonorrhoeae* vaccine antigens [47]. In this study, one protein sequence was described as type IV pilus secretin PilQ, present with over 80% identity. In *Pseudomonas aeruginosa*, it can be used as a new type of immunogen [48]. PilQ/PilA (QA) recombinant chimeric antigen is an effective candidate to prevent *P. aeruginosa* infection [49]. In *Legionella pneumophila*, PAL/PilE /FlaA DNA vaccine can stimulate strong humoral and cellular immune responses [50]. Five protein sequences described as OmpA proteins or OmpA family proteins were screened. Through sequence alignment, it was found that the OmpA protein widely existed in *Aeromonas* bacteria, and the homology among different genotypes of *A. hydrophila* was more than 80%. It was speculated that OmpA protein of *A. hydrophila* was a cross-reactive common antigen, which could cause protective immunity in fish. Previous studies have shown that OmpA1 can be used as a potential vaccine antigen against the infection of virulent *A. hydrophila* in channel catfish [12].

Candidate antigenic proteins were selected and prepared into recombinant subunit vaccines for immunization prevention, which has been widely used in aquaculture. But, the subunit vaccines usually contain only a single antigen, which leads to inefficient immunity, requiring multiple immunizations or the development of multiple vaccines based on multiple antigens [51]. The addition of the adjuvant is an effective way to enhance the immunoprotective effect of the vaccine [52]. Vaccine adjuvants can be divided into: aluminum adjuvant, mineral oil adjuvant, natural and synthetic polymers adjuvant, cytokine adjuvant, Toll-like receptors (Toll-like receptor, TLR) agonist adjuvant, liposome adjuvant, and nanomaterials adjuvant, etc. [53,54,55,56]. Researchers found significantly reduced mortality in the group challenged with *L. rohita* after immunization with modified adjuvant-based recombinant outer membrane protein R vaccine preparations when compared with mineral-oil-only and single-modified adjuvants [57]. Rauta et al. used polymeric carriers such as chitosan, poly-(D, l-lactide-co-glycolide) (PLGA), and polylactic acid (PLA) as an adjuvant coat for *A. hydrophila* Omp immune *L. rohita* and found a significant increase in specific antibody responses. It is speculated that PLA and PLGA fish non-intestinal immune new antigen carriers can not only replace free adjuvants to reduce side effects but also provide long-lasting immunity [58].

Individual proteins contain hundreds of epitopes, but not all epitopes can induce T-cell- and B-cell-mediated immune responses. Epitope vaccines are developed and designed based on pathogen epitopes or amino acid sequence characteristics of determining clusters. It has the advantages of low toxicity, high immune targeting, safe and stable removal of antigen except immune-related components and harmful parts, and can improve the immune protection effect through the combination of multiple epitopes [59]. However, peptides are weakly immunogenic, and in practical applications, optimized designs are needed to enhance the immune activity of epitope vaccines. Connecting multiple different epitopes to construct a multi-epitope vaccine (MEV) can effectively enhance the immunogenicity of the vaccine. Zhu et al. selected three antigens from the proteome of six *Nocardia* subspecies for epitope identification based on reverse vaccinology. The shortlisted T and B cell epitopes were fused with appropriate adjuvants and adaptors to construct the vaccine, and the immune simulation results suggest that the vaccine has the potential to induce a strong protective immune response in the host [60]. Li et al. analyzed the epitopes of *Brucella* outer membrane protein 10, outer membrane protein 25, outer membrane protein 31, and BtpB based on reverse vaccinology, and constructed a final MEV containing 806 amino acids by connecting adaptors and adjuvants, and found MEV to be highly immunogenic in animal experiments [61]. Sheng et al. prepared recombinant pyruvate dehydrogenase E1 subunit alpha (PDHA 1) and glyceraldehyde-3-phosphate dehydrogenase (GAPDH) subunit vaccines and found that both vaccines were capable of inducing specific antibody responses, and produced higher rates of immune protection than inactivated vaccines. Thus, they analyzed and identified the linear B cell epitopes of GAPDH and PDHA 1, and prepare a multi epitope vaccine, which can resist Streptococcus iniae infection after immunization with *Paralichthys olivaceus* [62]. Moreover, the peptide molecule can be modified to prevent new epitopes generated at the junction when the epitope is in tandem to affect the immune activity of the original epitope. When constructing a multi epitope vaccine against *Staphylococcus aureus*, Nawaz et al. first improved the vaccine’s ring refinement and structural stability through disulfide bond engineering transformation, and optimized it according to the usage mode of the *E. coli* (K12) expression system to achieve effective expression. Only the first three complexes docked with major histocompatibility complex class I (MHC-I), major histocompatibility complex class II (MHC-II), and TLR-4 receptors were selected as effective vaccine candidates [63]. To enhance the effectiveness of peptide vaccines, adjuvants are sometimes used along with peptide vaccines. Shafaghi et al. attached domain 4 of the candidate adjuvant to the end of the construct to enhance the immunogenicity of the epitope vaccine [64]. In conclusion, it is possible to design a multi-epitope-based subunit vaccine, which has better protection against multiple serotypes of *A. hydrophila*. Firstly, the epitope sequences of all selected antigens are predicted, calculated, and screened to determine their immunogenicity, toxicity, sensitization, and solubility. Then, adjuvants and ligands are combined with these sequences to effectively enhance the immune response. Next, the vaccine construction is modeled and docking and molecular dynamics simulations are used to predict the binding affinity of the designed vaccine with different immune receptors. Finally, the stability of the molecule is predicted by estimating the binding free energy of the complex. On this basis, Islam et al. have designed a multi epitope subunit vaccine for *A. veronii*, but the proposed candidate vaccine still needs to be tested in vivo and in vitro to determine its effectiveness and safety [65].

## 5. Conclusions

In this study, 39 outer membrane proteins were selected as vaccine candidate antigens using reverse vaccinology. These candidate proteins were all present in *A. hydrophila* and *Aeromonas* and showed high protein homology. A total of 16 proteins showed good conservation and they are described as OmpA-, OmpK-, and TonB-dependent transferrin family receptor, outer membrane protein transporter, TolC family outer membrane protein, peptidoglycan DD-metalopeptidase family protein, TIGR04219 family outer membrane barrel protein, immunosuppressant A, and DUF2860 family protein. These results will provide a wealth of candidate antigen materials for subsequent preparation.

## Figures and Tables

**Figure 1 vaccines-11-01266-f001:**
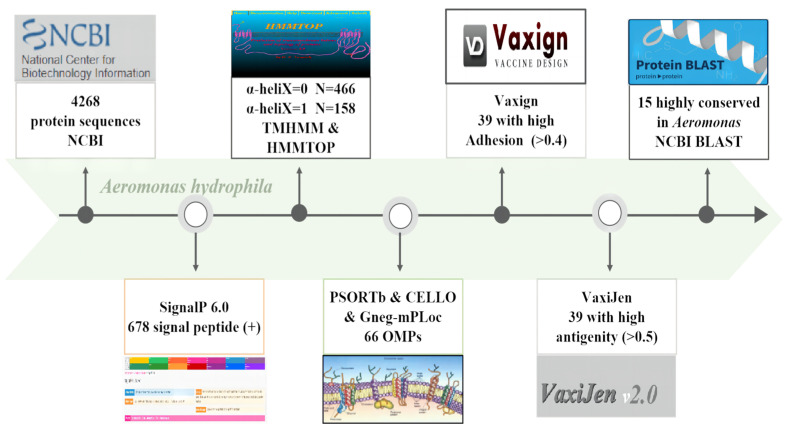
Schematic representation of the bioinformatics analysis of the *A. hydrophila* J-1 strain genome. Note: The whole genome nucleotide acid coding sequences of the *A. hydrophila* J-1 strain were downloaded in the FASTA file format from the NCBI database for a total of 4268 sequences (accession number: CP006883.1). The proteins with the signal peptide were predicted using SignalP 5.0; TMHMM 2.0 and HMMTOP were used to predict the transmembrane helical structures; and PSORTb, CELLO, and Gneg-mPLoc were used to predict the subcellular localization of the proteins. The potential antigenicity of the proteins was assessed using VaxiJe with a cutoff value set to 0.4; the adhesion of the proteins was assessed using Vaxign with an adhesion index greater than 0.5; and BLAST sequence alignment was used to assess protein homology in the *Aeromonas* sp.

**Figure 2 vaccines-11-01266-f002:**
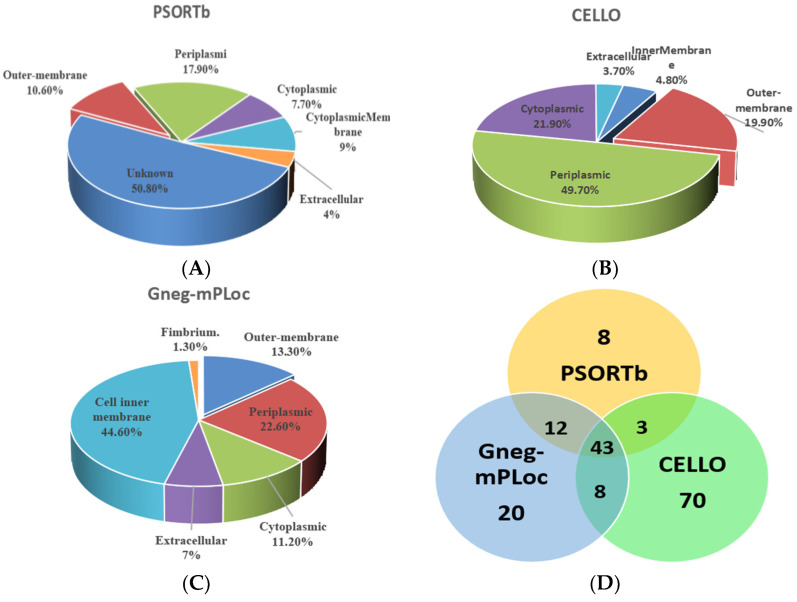
The subcellular localization prediction of candidate proteins. Note: (**A**): the result predicted by PSORTb; (**B**): the result predicted by CELLO; (**C**): the result predicted by Gneg-mPLoc; and (**D**): the cross-predicted results of the three software.

**Figure 3 vaccines-11-01266-f003:**
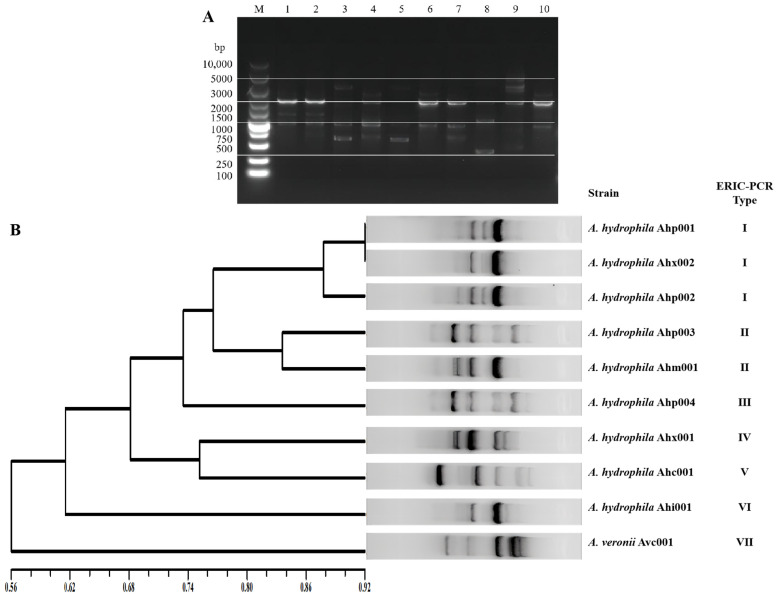
ERIC-PCR fingerprints and UPGMA clustering of 10 strains. Note: (**A**): ERIC-PCR fingerprints of 10 strains. Lanes: M, 10,000 bp DNA molecular size marker; 1, Ahp001; 2, Ahp002; 3, Ahp003; 4, Ahx001; 5, Ahp004; 6, Ahi001; 7, Ahm001; 8, Ahc001; 9, Avc001; 10, Ahx002. (**B**): Result of ERIC-PCR genotyping and UPGMA clustering of 10 strains. The clustering tree is shown on the left, the ERIC-PCR electropherogram in the middle, and the typing results on the right.

**Figure 4 vaccines-11-01266-f004:**
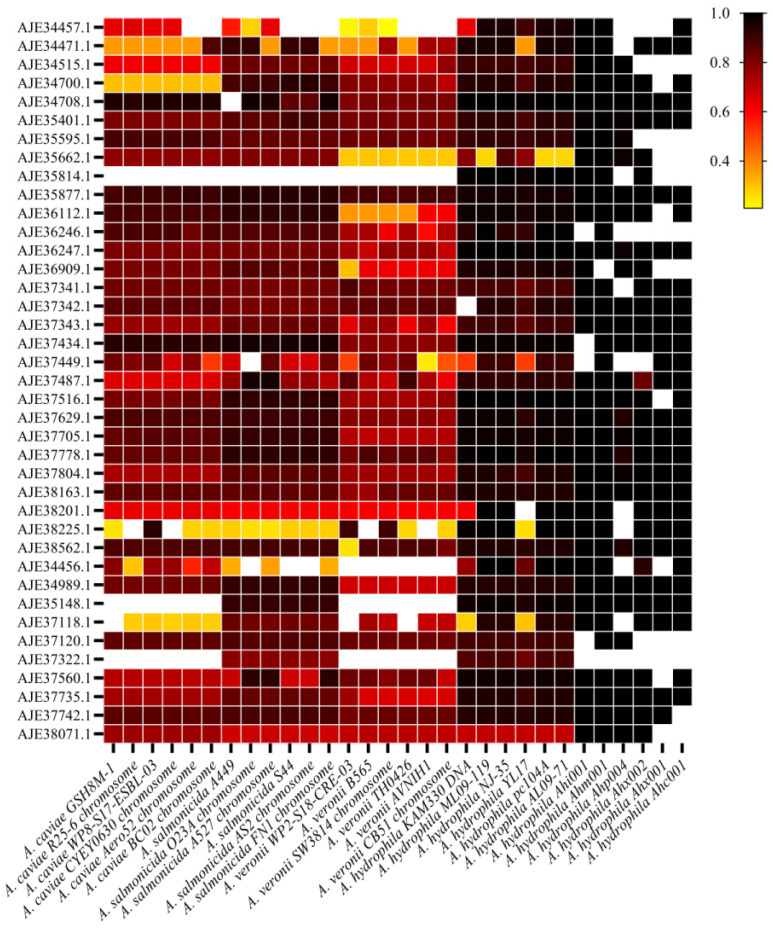
Heatmap showing the homology and distribution of 39 outer membrane protein sequences of 24 *Aeromonas* and six sequenced strains in the NCBI database. Note: The strain name is displayed at the bottom, and the sequence protein name is displayed on the left. The threshold for the presence of proteins in 30 strains is expressed as homogeneity. Yellow indicates an identity at around 20%, red at around 60%, and black at 100%. The darker the color, the higher the homogeneity. Blank indicates that the outer membrane protein sequence does not exist or the protein sequence coverage is less than 80%.

**Table 1 vaccines-11-01266-t001:** Information of *A. hydrophila* strains used in this study.

Strain	Host Fish Species	Isolated Tissues	Location
*A. hydrophila* Ahp001	*Parabramis pekinensis*	kidney	Nanjing
*A. hydrophila* Ahp002	*Parabramis pekinensis*	liver	Nanjing
*A. hydrophila* Ahp003	*Parabramis pekinensis*	gill	Huai’an
*A. hydrophila* Ahp004	*Parabramis pekinensis*	liver	Huai’an
*A. hydrophila* Ahm001	*Megalobrama amblycephala*	liver	Wuhan
*A. hydrophila* Ahc001	*Carassius auratus gibelio*	body surface	Hubei
*A. hydrophila* Ahi001	*Ictalurus punctatus*	kidney	Sichuan
*A. hydrophila* Ahx001	*Ictalurus punctatus*	body surface	Sichuan
*A. hydrophila* Ahx002	*Ictalurus punctatus*	body surface	Sichuan
*A. veronii* Avc001	*Carassius auratus*	gill	Lianyungang

**Table 2 vaccines-11-01266-t002:** Identification of candidate proteins for the development of subunit vaccines in the *Aeromonas* genus with high identity and occurrence rates.

Protein Number	Protein Description	α-Helix	Adhesion	Antigenicity	Percentage of Occurrence (%)
AJE34457.1	fimbrial biogenesis outer membrane usher protein	0	0.715	0.6061	8
AJE34471.1	TonB-dependent receptor	0	0.555	0.6085	38
AJE34515.1	porin OmpA	0	0.715	0.6038	16
AJE34700.1	ligand-gated channel protein	0	0.616	0.6769	30
AJE34708.1	outer membrane beta-barrel protein	0	0.838	0.7068	30
AJE35401.1	type IV pilus secretin PilQ	0	0.565	0.6985	57
AJE35595.1	carbohydrate porin	0	0.663	0.7585	51
AJE35662.1	efflux transporter outer membrane subunit	0	0.539	0.6502	32
AJE35814.1	MtrB/PioB family decaheme-associated outer membrane protein	0	0.723	0.5953	11
AJE35877.1	TolC family outer membrane protein	0	0.535	0.6247	70
AJE36112.1	siderophore amonabactin TonB-dependent receptor	0	0.657	0.6506	41
AJE36246.1	OmpP1/FadL family transporter	0	0.71	0.5389	22
AJE36247.1	outer membrane protein transport protein	0	0.825	0.6604	11
AJE36909.1	TonB-dependent hemoglobin/transferrin/lactoferrin family receptor	0	0.525	0.7156	14
AJE37341.1	OmpA family protein	0	0.612	0.725	19
AJE37342.1	porin OmpA	0	0.625	0.6875	54
AJE37343.1	OmpA family protein	0	0.559	0.6993	24
AJE37434.1	outer membrane protein assembly factor BamA	0	0.637	0.6605	49
AJE37449.1	maltoporin LamB	0	0.774	0.6319	43
AJE37487.1	outer membrane protein OmpK	0	0.726	0.5222	41
AJE37516.1	outer membrane beta-barrel protein	0	0.764	0.6778	11
AJE37629.1	TonB-dependent hemoglobin/transferrin/lactoferrin family receptor	1	0.624	0.665	43
AJE37705.1	LPS assembly protein LptD	1	0.527	0.7443	32
AJE37778.1	porin	0	0.704	0.7588	19
AJE37804.1	peptidoglycan DD-metalloendopeptidase family protein	0	0.787	0.7381	14
AJE38163.1	porin	0	0.767	0.7306	54
AJE38201.1	OmpA family protein	0	0.611	0.7071	3
AJE38225.1	porin	0	0.811	0.748	16
AJE38562.1	TonB-dependent siderophore receptor	0	0.591	0.5896	62
AJE34456.1	hypothetical protein	1	0.667	0.5206	5
AJE34989.1	TIGR04219 family outer membrane beta-barrel protein	0	0.691	0.7105	24
AJE35148.1	immune inhibitor A	0	0.572	0.5513	41
AJE37118.1	DUF1566 domain-containing protein	0	0.605	0.6726	5
AJE37120.1	outer membrane protein assembly factor BamE	0	0.608	0.6881	46
AJE37322.1	M23 family metallopeptidase	0	0.74	0.5346	11
AJE37560.1	OprD family outer membrane porin	0	0.644	0.6276	19
AJE37735.1	DUF2860 family protein	0	0.673	0.5925	16
AJE37742.1	outer membrane protein assembly factor BamC	0	0.643	0.6185	57
AJE38071.1	glycine zipper 2TM domain-containing protein	0	0.815	0.8437	57

Note: The number of transmembrane α-heliX of 39 proteins, adhesion index, and antigenicity. The occurrence rate of proteins with over 85% identity in protein sequences in 37 genera of *Aeromonas* sp.

**Table 3 vaccines-11-01266-t003:** Percentage of occurrence of 39 proteins in 30 *Aeromonas* isolates (%).

No.	Protein Number	NCBI NR Data Base				Sample Strains
		*A. caviae*	*A. salmonicida*	*A. veronii*	*A. hydrophila*	
1	AJE34457.1	67	50	50	100	50
2	AJE34471.1	100	100	100	100	83
3	AJE34515.1	100	100	100	100	50
4	AJE34700.1	100	100	100	100	83
5	AJE34708.1	100	83	100	100	100
6	AJE35401.1	100	100	100	100	100
7	AJE35595.1	100	100	100	100	50
8	AJE35662.1	100	100	100	100	67
9	AJE35814.1	0	0	0	100	50
10	AJE35877.1	100	100	100	100	100
11	AJE36112.1	100	100	100	100	83
12	AJE36246.1	100	100	100	100	17
13	AJE36247.1	100	100	100	100	100
14	AJE36909.1	100	100	100	100	50
15	AJE37341.1	100	100	100	100	83
16	AJE37342.1	100	100	100	83	100
17	AJE37343.1	100	100	100	100	100
18	AJE37434.1	100	100	100	100	83
19	AJE37449.1	100	83	100	100	50
20	AJE37487.1	100	100	100	100	100
21	AJE37516.1	100	100	100	100	83
22	AJE37629.1	100	100	100	100	100
23	AJE37705.1	100	100	100	100	100
24	AJE37778.1	100	100	100	100	100
25	AJE37804.1	100	100	100	100	100
26	AJE38163.1	100	100	100	100	100
27	AJE38201.1	100	100	100	83	83
28	AJE38225.1	67	100	67	100	83
29	AJE38562.1	100	100	100	100	100
30	AJE34456.1	100	50	0	100	67
31	AJE34989.1	100	100	100	100	100
32	AJE35148.1	0	100	0	100	100
33	AJE37118.1	83	100	67	100	83
34	AJE37120.1	100	100	100	100	33
35	AJE37322.1	0	100	0	100	0
36	AJE37560.1	100	100	100	100	83
37	AJE37735.1	100	100	100	100	100
38	AJE37742.1	100	100	100	100	83
39	AJE38071.1	100	100	100	100	67

Note: Numerical value indicates the proportion of the 39 protein positives present in the six strains of each species.

## Data Availability

All data generated or analyzed during this study are included in this article and its Appendix A. Further enquiries can be directed to the corresponding author.

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
