# Peer review of "The Screening of the Protective Antigens of Aeromonas hydrophila Using the Reverse Vaccinology Approach: Potential Candidates for Subunit Vaccine Development"

_vaccines, 2023, doi:10.3390/vaccines11071266_

Round 1

Reviewer 1 Report

This manuscript have screened the protective antigens of Aeromonas hydrophila via reverse vaccinology approach, and found two proteins were highly conserved in Aeromonas species, which could be selected as candidate antigens for vaccines development. This work may provide basic foundations for developing broad-spectrum protective vaccines of A. hydrophila. Overall, this paper is well written, concise in its content, and shows compelling data, the work has certain practicability. However, several issues need to be addressed:

1. In part 2.3.1., the number and name of the species described are inconsistent with those in the Table 1.

2. Only the ERIC-PCR analysis part of the paper is experimental data, but it seems to have little to do with the topic of the paper.

3. Although the two predicted target proteins are valuable, there is a lack of experimental verification, such as expression of their recombinant proteins, and immunization of mice to detect their serological response, neutralization effect in vitro, and animal protection effect. The experimental data for its verification should be increased so that the research can fully show its value and correspond to the level of this journal.

Author Response

On behalf of my co-author, we are very grateful to you for giving us an opportunity to revise our manuscript entitled “Screening of protective antigens of Aeromonas hydrophila via reverse vaccinology approach: potential candidates for subunit vaccines development" (Manuscript ID: vaccines-2481916 ). Those comments are all valuable and very helpful for revising and improving our paper. We have carefully studied reviewers’ comments and tried our best to revise our manuscript according to these comments. The following are the responses and revisions that we have made in response to the reviewers’ questions and suggestions on an item-by-item basis. Thanks again to the hard work of the editor and reviewer!

Point 1: In part 2.3.1., the number and name of the species described are inconsistent with those in the Table 1.

Response 1: Thank you for your suggestions. we have examined and revised the number and names of species in Table 1 and the manuscript.

Point 2: Only the ERIC-PCR analysis part of the paper is experimental data, but it seems to have little to do with the topic of the paper.

Response 2: Thank you for your comments. As we know, there are many different serotypes and genotypes of Aeromonas hydrophila, thereby the ERIC-PCR analysis was performed to confirmed the genotypes of A. hydrophila that isolated from diseased fish. Then, the genomic DNA of A. hydrophila with different genotypes  were extracted, and the ORF regions of identified 39 candidate antigens were amplified by PCR technology, which were then sequenced and the homology of these antigens were analyzed. In a word, the purpose of genotyping (ERIC-PCR analysis) and homology analysis are screening and validation of highly conserved antigens from the highly heterogeneous A. hydrophila strains, thereby laying the foundation for the development of broad-spectrum protective subunit vaccines for A. hydrophila.

Point 3: Although the two predicted target proteins are valuable, there is a lack of experimental verification, such as expression of their recombinant proteins, and immunization of mice to detect their serological response, neutralization effect in vitro, and animal protection effect. The experimental data for its verification should be increased so that the research can fully show its value and correspond to the level of this journal.

Response 3: Thank you for your suggestions. We appreciate and agree with your suggestions, we would like to prepare the recombinant proteins and verify their immune protective effects one by one, while this part of the work takes a lot of time, thus can not be completed in a short time. Previously, we have assessed the immune protective effect of OmpA subunit vaccine, which has been published in Frontiers in Immunology: “Immunogenicity and protective efficacy of OmpA subunit vaccine against Aeromonas hydrophila infection in Megalobrama amblycephala: An effective alternative to the inactivated vaccine”. In addition, we are detecting the immune protective effects of other candidate antigens, such as OmpTS, Aha1, TolC, TonB and the like, which maybe published in the future.

Minying Zhang, Ting Zhang, Yang He, Hujun Cui, Hong Li, Zehua Xu, Xu Wang, Yunlong Liu, Hongping Li, Xiaoheng Zhao, Hanliang Cheng, Jianhe Xu, Xiangning Chen, Zhujin Ding*. Immunogenicity and protective efficacy of OmpA subunit vaccine against Aeromonas hydrophila infection in Megalobrama amblycephala: An effective alternative to the inactivated vaccine. Frontiers in Immunology, 2023, 14: 1133742. doi: 10.3389/fimmu.2023.1133742.

Reviewer 2 Report

The manuscript presented t screening of protective antigens of Aeromonas hydrophila for subunit vaccines development. Before the paper could be accepted for publication in Vaccines, the authors need make revisions as follow:

1. Aeromonas hydrophila or A. hydrophila should be italic., please unify them.

2. Type IV pilus secretin PilQ 30 (AJE35401.1), TolC family outer membrane protein (AJE35877.1) were highly conserved in Aeromonas species, which could be selected as candidate antigens for vaccines development. AJE35401.1 and AJE35877.1 were membrane protein and as linear epitope, how to keep the immunogenicity as candidate antigens? Please discuss the strategy.

Quality of English language could be polished further.

Author Response

On behalf of my co-author, we are very grateful to you for giving us an opportunity to revise our manuscript entitled “Screening of protective antigens of Aeromonas hydrophila via reverse vaccinology approach: potential candidates for subunit vaccines development" (Manuscript ID: vaccines-2481916 ). Those comments are all valuable and very helpful for revising and improving our paper. We have carefully studied reviewers’ comments and tried our best to revise our manuscript according to these comments. The following are the responses and revisions that we have made in response to the reviewers’ questions and suggestions on an item-by-item basis. Thanks again to the hard work of the editor and reviewer!

Point 1: Aeromonas hydrophila or A. hydrophila should be italic., please unify them.

Response 1: Thank you for your suggestions. We re-examined and revised the font format included in the manuscript.

Point 2: Type IV pilus secretin PilQ 30 (AJE35401.1), TolC family outer membrane protein (AJE35877.1) were highly conserved in Aeromonas species, which could be selected as candidate antigens for vaccines development. AJE35401.1 and AJE35877.1 were membrane protein and as linear epitope, how to keep the immunogenicity as candidate antigens? Please discuss the strategy.

Response 2: Thank you for your comments. According to your suggestions, at the end of the discussion section, we have discussed how the linear epitope remains immunogenicity as a candidate antigen.

Reviewer 3 Report

The article by Ting Zhang et al. shows the preliminary results of antigenicity determination of different strains of Aeromonas spp. with a view to developing a vaccine against A. hydrophila infection.

Minor revisions are required.

Numbers less than or equal to 12 should be written out in full, bacterial names should be italicized.

Software/databases should be appropriately referenced in the bibliography section of the article.

Line 176 Bi et al. 2007 must be properly referenced like the other references.

In section 2.3.2, the authors should provide more information on the PCR kit used (manufacturer, city, country).

Line 182: how were the sequences produced?

Paragraph 3.1.1. the first lines and figure 1 should be in the method section.

Figure 2: Figures are difficult to read (3D).

Figure 3: Indicate the bands of interest.

Author Response

On behalf of my co-author, we are very grateful to you for giving us an opportunity to revise our manuscript entitled “Screening of protective antigens of Aeromonas hydrophila via reverse vaccinology approach: potential candidates for subunit vaccines development" (Manuscript ID: vaccines-2481916 ). Those comments are all valuable and very helpful for revising and improving our paper. We have carefully studied reviewers’ comments and tried our best to revise our manuscript according to these comments. The following are the responses and revisions that we have made in response to the reviewers’ questions and suggestions on an item-by-item basis. Thanks again to the hard work of the editor and reviewer!

Point 1: Numbers less than or equal to 12 should be written out in full, bacterial names should be italicized.

Response 1: Thanks for pointing out the errors. We have re-examined and revised the font format included in the manuscript.

Point 2: Software/databases should be appropriately referenced in the bibliography section of the article.

Response 2: Thank you for your suggestions. The software / databases have been cited in the Bibliography section.

Point 3: Line 176 Bi et al. 2007 must be properly referenced like the other references.

Response 3: Thank you, we have corrected it.

Point 4: In section 2.3.2, the authors should provide more information on the PCR kit used (manufacturer, city, country).

Response 4: Thank you for your suggestions. We have added information about the manufacturers, cities, and countries of the PCR kits in the manuscript.

Point 5: Line 182: how were the sequences produced?

Response 5: Thank you for your suggestions. The 16S rDNA gene and ERIC universal primer sequences for the parts of 2.3.2. and 2.3.3. were obtained from References. We have added the corresponding references.

Point 6: Paragraph 3.1.1. the first lines and figure 1 should be in the method section.

Response 6: Thank you for your suggestions. We have adjusted in the manuscript.

Point 7: Figure 2: Figures are difficult to read (3D).

Response 7: Thank you for your suggestions. We have revised the description of Figure 2 to improve the readability.

Point 8: Figure 3: Indicate the bands of interest.

Response 8: Thank you for your suggestions. We have added reference lines to point out the important bands in Figure 3.

Round 2

Reviewer 1 Report

Except for some experiments suggested to be supplemented, which the author said could not be supplemented in the short term, the remaining issues have been addressed by the authors.